# Association Between Amino Acid Polymorphisms in MICA and MICA-NKG2D Interaction Affinity: Implications and Significance for Tumor Immunity

**DOI:** 10.3390/biom16010047

**Published:** 2025-12-28

**Authors:** Chuyu Xiao, Wang Wang, Yangyang Zhang, Ting Huang, Chunjing Chen, Biyuan Liu, Chang Liu, Yingying Yang, Fangguo Lu, Quan Zhu

**Affiliations:** 1Department of Immunology, School of Medicine, Hunan University of Chinese Medicine, Changsha 410208, China; 202408020133@stu.hnucm.edu.cn (C.X.); wang18038882005@stu.hnucm.edu.cn (W.W.); 202308010839@stu.hnucm.edu.cn (Y.Z.); 004205@hnucm.edu.cn (B.L.); winlyl@stu.hnucm.edu.cn (Y.Y.); 2Department of Pathology, School of Medicine, Hunan University of Chinese Medicine, Changsha 410208, China; 004929@hnucm.edu.cn; 3Department of Pathogenic Biology, School of Medicine, Hunan University of Chinese Medicine, Changsha 410208, China; 004787@hnucm.edu.cn; 4Department of Histology and Embryology, School of Medicine, Hunan University of Chinese Medicine, Changsha 410208, China; liuchang004273@hnucm.edu.cn

**Keywords:** MICA, polymorphism, tumor immunity, NK cells

## Abstract

Major histocompatibility complex class I-like related gene A (MICA) is the most polymorphic non-classical *HLA* gene. MICA proteins are expressed at low levels on the surface of normal cells but are highly expressed on the surface of tumor cells. Its most important biological function is to bind to activating receptors on the surface of natural killer (NK) cells or CD8+ T cells, then activate these immune cells to exert immune killing effects. Multiple studies have shown that the amino acids at specific loci in the MICA molecule can significantly affect its binding ability to NKG2D. The binding strength of MICA-NKG2D significantly affects the anti-tumor effect of NK cells in the body and the prognosis of many tumor patients. However, the strong MICA-NKG2D interaction can trigger negative feedback against this immune response by down-regulating the expression of NKG2D or generating soluble MICA, weakening the overly intense immune response. Therefore, simply evaluating the intensity of the anti-tumor immune response from the perspective of the amino acid polymorphism of MICA affecting its binding ability to NKG2D also has limitations. We review the effects of MICA amino acid polymorphism on the affinity of the NKG2D signal pathway and analyze in detail the specific role of MICA amino acid polymorphism in tumor immunity. The study provides a reference for understanding the mechanism of anti-tumor immune response by NK cells or other immune cells, as well as a theoretical basis for considering the MICA-NKG2D signal axis for anti-tumor immune therapy in future clinical practice.

## 1. Introduction

The major histocompatibility complex (*MHC*) gene cluster is located on the short arm of human chromosome 6, and all genes that it encodes are closely related to the immune response and immune regulation. In humans, the MHC is also known as human leukocyte antigen (HLA). The classical HLA-I and HLA-II molecules can present antigens and participate in the adaptive immune response, mediated by T and B cells [1,2,3]. Thus, they are crucial members of the *HLA* gene cluster. Major histocompatibility complex class I-like genes (MIC) belong to the family of non-classical *HLA-I* genes. The *MIC* gene family has seven members, but only *MICA* and *MICB* genes can encode functional proteins; the remainder are pseudogenes [4,5]. MICA and MICB proteins are highly similar to HLA-I molecules and have α1, α2, and α3 domains. However, they lack β2 microglobulin and cannot present antigens. Physiologically, MICA and MICB are expressed at low levels in epithelial cells and vascular endothelial cells [6,7], and they are almost unexpressed in other cells. When cells are exposed to stress, such as a viral infection or tumor transformation, MICA and MICB will be highly expressed on their surfaces, serving as immune markers [8,9,10].

Since both *MICA* and *MICB* genes are located within the human *MHC* gene cluster, as with other *HLA* genes, they also possess high levels of polymorphism. As of 11 October 2025, 589 *MICA* and 312 *MICB* alleles have been recorded in the IMGT/HLA database (https://www.ebi.ac.uk/ipd/imgt/hla/, accessed on 11 October 2025). These alleles encode 279 MICA and 53 MICB molecules, respectively. High polymorphism is an important characteristic of *HLA* genes. The distribution characteristics of *MICA* and *MICB* genes vary significantly among different populations or disease groups. Numerous studies have demonstrated that certain *HLA* gene polymorphisms are strongly correlated with susceptibility and prognosis in multiple diseases [11,12,13]. *MICA* and *MICB* gene polymorphisms are also reported to be associated with the occurrence and development of tumors [14], viral infections [15,16], and autoimmune diseases [17,18]. Therefore, their polymorphisms have significant immunological significance.

Beyond MICA and MICB, ULBP1-6 are ligands of NKG2D [19,20]. However, since the genes encoding ULBP1-6 are not located in the MHC gene cluster on human chromosome 6, the ULBP family is highly conserved [21].

From the perspective of biological function, MICA mainly acts as a ligand for the activating receptor NKG2D on the surfaces of various immune cells, such as natural killer (NK) cells, CD8+ T cells, NKT cells, and γδT cells, and enables them to exert their killing effects [22,23,24,25,26]. The two can combine and subsequently activate NK cells, enabling immune surveillance [27]. Previous studies have shown that genetic polymorphisms of MICA are statistically associated with the incidence of various cancers [14,28], but the molecular mechanism behind this has not been clarified. Recently, a study has elucidated the causal relationship between MICA amino acid polymorphisms and the degree of activation of NK cells from the perspective of amino acid polymorphisms at specific amino acid sites in MICA molecules, explaining the molecular mechanism underlying the relationship between *MICA* gene polymorphisms and disease susceptibility [29]. Most epithelial-derived tumor cells, as well as some non-epithelial-derived tumor cells (such as leukemia, melanoma, etc.; see Table 1), express high levels of MICA and MICB, which can attract immune cells expressing NKG2D and enable them to exert their killing effects [22,23,24,25,26]. However, MICA or MICB polymorphic molecules exhibit significant differences in their abilities to activate NK cells and other immune cells [30]. These differences are closely related to the degrees of tumor progression, immune escape, and activation of immune cells. Ligands such as MICA, when highly expressed on the surfaces of tumor cells, can be regarded as immune checkpoints. After binding to NKG2D, MICA can effectively activate NK cells and CD8+ T cells to exert anti-tumor effects [27,31]. At present, the majority of research on NKG2D ligands focuses on MICA. As a result, it is known that the intensity of the interaction between MICA and NKG2D determines the regulatory strength of this immune checkpoint regarding the immune effects of the body. According to relevant studies, NKG2D is a highly conserved cell membrane surface protein, and genetic and amino acid polymorphisms of NKG2D are rare [32]. Amino acid polymorphisms of MICA can be regarded as key factors determining the intensity of the interaction between MICA and NKG2D. Therefore, it is crucial to achieve a comprehensive understanding of the influence of MICA amino acid polymorphisms on the interaction between MICA and NKG2D to strengthen anti-tumor immunity and improve the prognoses and outcomes of patients. This would also provide a theoretical basis for the use of the MICA-NKG2D signaling axis for treatment in the field of tumor immunology.

This article reviews and summarizes existing research on the roles of *MICA* gene or amino acid polymorphisms in tumor immunity, systematically discusses the influence of MICA amino acid polymorphisms on the strength of the MICA-NKG2D interaction, and elaborates on the negative feedback mechanism that the immune system generates in response to overly strong MICA-NKG2D interactions. These aspects are comprehensively considered in regard to the immune evasion of tumor cells targeting the MICA-NKG2D signaling axis, enabling an in-depth understanding of MICA/B amino acid polymorphisms and also offering new perspectives for the application of the MICA-NKG2D immune checkpoint to enhance anti-tumor immunity in the future.

## 2. The Ability of MICA Polymorphic Molecules to Activate NK Cells Is Closely Related to the Types of Amino Acids at Their Specific Sites

### 2.1. The Amino Acid Sequences of All MICA Polymorphic Molecules Contain Multiple Pairs of Sites with Opposing Co-Occurrence Patterns

Since the *MICA* gene is located in the human *HLA* gene region and belongs to the human major histocompatibility complex gene cluster, the gene and its protein have abundant polymorphisms. According to the IMGT/HLA database, the amino acid sequences of these MICA polymorphic molecules contain a total of 27 amino acid polymorphism sites. Comparing the amino acid polymorphisms at these sites reveals that there are multiple sets of co-occurrence patterns. The most significant finding is that, at positions 36, 129, 173, 206, 210, and 215, the amino acids show two opposing co-occurrence patterns. Some of the MICA polymorphic molecules (e.g., MICA*001) at these six sites show a co-occurrence pattern of C-M-K-G-W-S, while others (e.g., MICA*004) show a co-occurrence pattern of Y-V-E-S-R-T. Only a few MICA molecules (such as MICA*028 and MICA*042) do not follow the above co-occurrence patterns.

Based on the amino acid types at the above six loci, the MICA polymorphic molecules can be classified into three types: those that present the C-M-K-G-W-S pattern (represented by MICA*001), those that present the Y-V-E-S-R-T pattern (represented by MICA*004), and those that are non-strictly linked (represented by MICA*028). In addition, there are more detailed distinctions within these categories. Under the premise that all six loci display the pattern C-M-K-G-W-S, the W/G diploidy at the 14th amino acid locus can be used to further divide these MICA polymorphic molecules into two subtypes. Some molecules (such as MICA*001, MICA*007, and MICA*012) show a co-occurrence pattern of W-C-M-K-G-W-S at the above-mentioned loci, while others (such as MICA*002, MICA*011, and MICA*015) show the pattern G-C-M-K-G-W-S. On the other hand, under the premise that all six loci exhibit the pattern Y-V-E-S-R-T, the L/V, G/S, and Q/R diploidies at the 122nd, 175th, and 251st amino acid loci can be used to further divide these molecules into three subtypes. Some molecules (e.g., MICA*004) show a co-occurrence pattern of Y-V-V-E-S-S-R-T-Q at the above-mentioned loci, while others (e.g., MICA*005) show the pattern Y-L-V-E-G-S-R-T-R. Moreover, a small number of molecules do not strictly follow the above-mentioned pattern in terms of the amino acid types at these loci. Please refer to Figure 1 for details.

The co-occurrence patterns of the amino acid loci of the MICA polymorphisms mentioned above provide an important basis for the classification of MICA molecules into subtypes. The types of amino acids at the sites of these co-occurrence patterns are closely related to differences in the biological functions of such MICA molecules.

### 2.2. The Amino Acid Co-Occurrence Patterns at Certain Sites Affect the MICA-NKG2D Signal Activation Intensity

The amino acid sequences of MICA molecules exhibit polymorphisms at multiple sites, and specific sites also show unique co-occurrence patterns. The effects of these characteristics on the biological functioning of MICA are of particular interest. MICA is the most important ligand of NKG2D, and its main function is to bind to NKG2D and activate NK cells, enabling them to kill target cells. A report by Steinle et al. 2001 [56] discussed whether MICA polymorphic molecules differ in their abilities to activate NKG2D. However, they only studied the M/V polymorphism at the 129th site of an MICA molecule. They found that, when the amino acid at this site was methionine (Met/M), the ability of MICA to activate NK cells was significantly enhanced; meanwhile, when the amino acid at this site was valine (Val/V), the opposite effect was observed [56]. Other studies support the above findings, suggesting that MICA with M at the 129th site has a strong ability to activate NK cells [57,58]. Previously, due to the insufficient understanding of MICA, the M/V polymorphism at the 129th amino acid site was regarded as the sole determinant of MICA’s ability to activate NK cells. See Figure 2 for details.

The question of whether amino acid polymorphisms at other sites could also affect the activation of NKG2D by MICA remained unresolved for a long time. In 2009, Zou et al. adsorbed and eluted sera from kidney transplant patients using MICA*001 and MICA*009 proteins and obtained two anti-MICA antibodies (named MICA-G1 and MICA-G2); they then incubated these antibodies with MICA*002 and MICA*008, respectively. It was found that the binding of MICA*002 and MICA*008 to the two anti-MICA antibodies showed contrasting characteristics: MICA*002 had a strong affinity for G1 (mean fluorescence intensity > 5000) and had a low affinity for G2 (mean fluorescence intensity < 1000), while MICA*008 had a strong affinity for G2 and had a low affinity for G1. For details, please refer to Ref. [59]. It should be noted that the 129th amino acid in both MICA*001 and MICA*002 is M, while in MICA*008 and MICA*009, it is V. As a result, there is significant variation in the affinities of these four MICA proteins for antibodies. In their work, Zou et al. used anti-MICA antibodies instead of the NKG2D protein to bind with MICA. However, since most anti-MICA antibodies have binding sites in the α1 and α2 domains [60], and the NKG2D receptor also combines with the MICA protein in these domains, the sequences of anti-MICA antibodies and NKG2D receptors that bind with MICA are highly similar. Subsequently, Zou et al. found that, when the 129th amino acid of MICA*008 was mutated from V to M, while other positions remained unchanged, the reaction of the MICA*008 mutant did not change. At the same time, they kept the 129th amino acid of MICA*002 unchanged and mutated multiple other amino acids, after which they found that the reaction of the MICA*002 mutant changed [59]. Although these studies focused on the affinities of antibodies to MICA, they suggest that the binding strength of MICA polymorphic molecules to antibodies or NKG2D receptors may not be solely related to the nature of the 129th amino acid, as reported by Steinle.

Luo et al. further contributed to this area of research. They selected 29 MICA proteins that all exhibited over 90% coverage across different populations and were highly representative. They found that there were significant differences in the affinities of these proteins for specific anti-MICA antibodies and NKG2D receptors. Based on the binding strength, the 29 studied proteins could be divided into two major types (named types I and II). Type I MICA molecules had significantly greater affinities for the NKG2D receptor, and their capacities to activate NK cells and engage in killing were also significantly stronger compared to type II molecules [29]. The reason for this lies in the differences in the amino acid types at specific sites between the two types of MICA molecules. The authors concluded that, in type I MICA molecules, the amino acids at the 36th, 129th, 173rd, 206th, 210th, and 215th sites showed a co-occurrence pattern of C-M-K-G-W-S, while the amino acids at the same six sites in type II MICA molecules showed a co-occurrence pattern of Y-V-E-S-R-T.

According to the IMGT/HLA database, we classified the MICA polymorphic molecules into three types based on the amino acid polymorphisms at the 36th, 129th, 173rd, 206th, 210th, and 215th sites. However, in the study by Luo et al., the affinities of MICA polymorphic molecules to NKG2D did not show three states. Some MICA polymorphic molecules with less rigid amino acid linkages at the six sites mentioned above (such as MICA*028) were also classified as either strongly binding or weakly binding types (i.e., type I and type II). Among these MICA polymorphic molecules, MICA*028, MICA*037, and MICA*042 are particularly noteworthy. In MICA*028, the amino acids at the above six sites are Y-V-E-G-W-S, while those in MICA*037 are C-M-K-S-R-T. Thus, type I and type II MICA molecules involve mixed-type linkages. However, the affinities of these two MICA molecules for NKG2D still appear to be related to the amino acid types at the 36th, 129th, and 173rd sites. MICA*037, with a C-M-K amino acid linkage, is a type I MICA molecule, while MICA*028, with Y-V-E, is classified as type II MICA. Moreover, MICA*042 has Y-M-K-S-R-T at the above six sites. Regarding MICA*042, although the 129th amino acid is M and the 173rd amino acid is also that of type I MICA, the amino acids at the other four sites are all those of type II MICA, and its affinity for NKG2D is relatively weak. Thus, it is classified as a type II MICA polymorphic molecule. This once again highlights that the M/V dimorphism at the 129th amino acid site is not the sole determinant of the strength of activation of NK cells, and other polymorphic sites also contribute.

Furthermore, Yang et al. conducted a computer simulation of all 5225 mutations of the MICA protein. It was shown that, apart from the M/V dimorphism at the 129th site, which can affect the affinity and stability of MICA-NKG2D, dimorphisms at multiple other positions also have similar effects. They discovered that the dimorphisms at positions 14, 36, 129, 173, and 206 had a significant impact on the MICA-NKG2D binding affinity. These five positions also exhibit the amino acid polymorphism co-occurrence patterns mentioned previously. When the amino acid at the 14th position changes from W to G, that at the 36th position changes from C to Y, that at the 129th position changes from M to V, and that at the 206th position changes from G to S, the stability of MICA-NKG2D is reduced. Only the 173rd position, where K is mutated to E, can promote the stability of MICA-NKG2D [61]. Thus, according to the classification of MICA polymorphic molecules proposed by Luo et al., the affinities of type I MICA for NKG2D are stronger than those of type II MICA. The reason for this may be that the amino acid polymorphisms at these positions affect the stability of MICA-NKG2D. Type I MICA contains most of the amino acids that are favorable for the binding of MICA-NKG2D, while type II MICA does not.

Steinle et al. initially proposed that the M/V polymorphism at the 129th amino acid site of MICA affected the activation of NK cells. However, they did not conduct in-depth research on the molecular mechanism behind this. Specifically, the ways in which the M/V polymorphism at the 129th site affected the activity of NK cells were not elaborated in their study. Zou et al. extended the work of Steinle et al. from the affinities of anti-MICA antibodies to MICA. They were the first to discover that amino acid polymorphisms at other sites could also affect the affinities of MICA and antibodies, demonstrating that the 129th amino acid was not the sole factor determining the affinity of MICA. However, Zou et al. did not use the NKG2D protein in their research. Although the regions where MICA interacts with antibodies or NKG2D receptors are located in the α1 and α2 domains, Zou et al. failed to identify any differences in affinity between the MICA polymorphic molecules and NKG2D. Lastly, Luo et al. clarified that the co-occurrence patterns of six specific positions in the amino acid sequence of MICA are the fundamental reasons for its affinity with NKG2D. They also proved that the amino acid polymorphisms at these six sites affected the intensity of activation of the NKG2D signaling pathway, thereby influencing the activity of NK cells.

## 3. The *MICA* Gene, Which Is Related to Tumor Susceptibility, Exhibits a Complex and Contradictory Set of Polymorphisms

As the most polymorphic non-classical *MHC* gene, MICA has long attracted the attention of researchers, particularly regarding its role in tumor pathogenesis. Polymorphisms of the *MICA* gene are closely related to susceptibility to various tumors. Based on the above, the amino acid polymorphisms at specific loci may affect the strength of the interaction between MICA and NKG2D. If the amino acids at specific loci are conducive to binding between MICA and NKG2D, this may facilitate the activation of NK cells and enable anti-tumor effects. Therefore, the probability of tumor occurrence in the population carrying the corresponding MICA allele should be lower. However, the reality does not fully align with this. In fact, the distribution of *MICA* gene polymorphisms in the tumor-affected population is complex and contradictory.

### 3.1. MICA Alleles That Carry the C-M-K-G-W-S Amino Acid Pattern Are Negatively Correlated with the Occurrence of Various Tumors

Given that MICA is mainly expressed on the surfaces of malignant tumor cells derived from epithelial cells, we sought to analyze the association between common epithelial cell-derived malignant tumors and MICA. Toledo-Stuardo et al. found that the survival rate of gastric cancer patients carrying MICA*008 and MICA*009 was significantly lower than that of patients carrying MICA*002. Meanwhile, the survival rate of gastric cancer patients carrying the homozygous allele of MICA*002 was significantly higher than that of patients carrying MICA*002/008 or MICA*002/009 [62]. Ouni’s research indicated that 71% of breast cancer patients under the age of 40 carried the Val/Val genotype at the 129th site of MICA, and most breast cancer patients with a family history also had this genotype. This suggests that the M/V polymorphism at the 129th site is closely related to the onset of breast cancer [63]. In Tani’s study, it was found that the number of oral cancer patients carrying the MICA-A5.1 genotype was significantly higher than that among healthy controls. Based on the association between the two MICA genotypes, most *MICA-A5.1* genes correspond to the MICA*008 allele [64]. Meanwhile, based on an analysis of amino acid polymorphic sites, MICA*002 presents a C-M-K-G-W-S co-occurrence pattern at sites 36, 129, 173, 206, 210, and 215, aligning with the type I MICA proposed by Luo et al. [29]. Moreover, MICA*008 and MICA*009 present a Y-V-E-S-R-T sequence at the above six sites, aligning with type II MICA. The former activates NK cells more strongly than the latter, so the intensity of activation of NK cells and CD8+ T cells in tumor patients carrying different *MICA* polymorphic genes is likely to vary significantly, and this is postulated to affect the body’s anti-tumor immune response.

In addition to solid tumors, hematological malignancies are also closely related to the MICA genotype. Luo et al. observed that the presence of MICA*010 in leukemia patients (including 32 acute myeloid leukemia patients, 46 chronic myeloid leukemia patients, 23 acute lymphocytic leukemia patients, and 6 chronic lymphocytic leukemia patients) from the Southern Han Chinese population was significantly higher than in the normal control group. After a ratio analysis (OR), it was determined that *MICA*010* was a susceptibility gene for leukemia [65]. Moreover, in the aforementioned work, the authors compared the amino acid sequences of MICA polymorphisms. It was found that, based on the co-occurrence pattern of C-M-K-G-W-S at the 36th, 129th, 173rd, 206th, 210th, and 215th positions, the W/G dimorphism of the 14th amino acid could be used to further divide the MICA polymorphic molecules into more specific subtypes. The report by Alena et al. expanded on this. They found that, in patients with acute myeloid leukemia, when the amino acid at the 14th position was glycine (G), the overall survival time of patients after hematopoietic stem cell transplantation was lower than that in the group with W at this position. This suggests that W/G dimorphism at the 14th position might also be a factor affecting the affinity of MICA and NKG2D [66]. However, the amino acid polymorphism at the 14th position is present in all MICA polymorphisms. Based on the co-occurrence pattern of Y-V-E-S-R-T at the 36th, 129th, 173rd, 206th, 210th, and 215th positions of the MICA protein, it is possible to achieve more detailed division using W/G. Therefore, the specific impact of the amino acid polymorphism at this position on the binding of MICA and NKG2D may be more complicated.

### 3.2. MICA Polymorphic Molecules That Carry the C-M-K-G-W-S Amino Acid Sequence Can Also Promote the Occurrence and Development of Tumors

According to Luo et al., MICA molecules that present C-M-K-G-W-S at the 36th, 129th, 173rd, 206th, 210th, and 215th sites can be defined as type I MICA, while molecules that present Y-V-E-S-R-T at these six sites can be defined as type II MICA [29]. There are significant differences in their abilities to activate the NKG2D signaling pathway. Type I MICA has a stronger ability to activate NK cells, and it can maintain immune surveillance and anti-tumor immunity to a greater extent than type II MICA. Thus, Luo et al. posit that individuals carrying type I MICA have better outcomes and treatment prognoses than those carrying type II MICA.

However, our review of the literature indicates that the type I MICA mentioned by Luo et al. does not always have a restrictive effect on malignant tumors. Ding et al. found that the proportion of colon cancer patients carrying the MICA*012 allele was significantly higher than in healthy individuals, while the proportions of patients carrying the *MICA*009* and *MICA*049* alleles were significantly lower. They also discovered that the overexpression of *MICA*012* in colon cancer cell lines could significantly promote the proliferation, invasion, and metastasis of colon cancer cells [67]. Antje et al. reported that, in melanoma cell lines without MICA expression, the overexpression of the two MICA isoforms MICA-129M and MICA-129V caused tumor cells to produce higher levels of soluble MICA (sMICA) [68]. sMICA can also bind to the NKG2D receptor, but it cannot activate the downstream signaling of NKG2D. It not only fails to activate NK cells, but it also occupies the site on the cell membrane where NKG2D binds to MICA, preventing immune cells such as NK cells from recognizing and killing tumor cells, which is an important step allowing tumor cells to evade NK cell killing [69,70,71,72,73,74]. In another study on polymorphisms of the *MICA* gene and nasopharyngeal carcinoma, it was also mentioned that the copy number of EB virus in the tumor tissue of nasopharyngeal carcinoma patients carrying MICA-129M was significantly higher than in patients carrying MICA-129V [75]. Moreover, in the treatment of various hematological malignancies with hematopoietic stem cell transplantation, researchers found that, when the *MICA* alleles of patients encoded 129-M homozygotes, the risk of recurrence significantly increased [76]. These reports all indicate that the factors influencing the strength of the affinity between MICA and NKG2D are highly complex. Moreover, the division of MICA into strongly binding and weakly binding types based solely on the amino acid types at specific positions has certain limitations.

In addition, it should be noted that MICA molecules exist in two forms in the human body, namely membrane-bound MICA and soluble MICA. The former is expressed on the cell surface and functions by binding to the NKG2D receptor, and its binding strength is influenced by the aforementioned type I and type II MICA. sMICA is formed via the shedding of membrane-bound MICA. This MICA can also bind to NKG2D, but, regardless of whether it is type I or type II sMICA, its binding to NKG2D cannot activate the downstream signaling pathway. Instead, it occupies the domain where NKG2D binds to the ligand molecules on the target cell surface, preventing NK cells’ activation. The reasons for the formation of sMICA are described in the subsequent section.

## 4. MICA, Which Has a Strong Affinity for NKG2D, Triggers a Negative Feedback Mechanism to Downregulate the Activation of NK Cells

### 4.1. MICA, Which Has a Strong Affinity for NKG2D, Can Cause the Internalization of the NKG2D Receptor on the Surfaces of Immune Cells

The classification of MICA types based on the affinity between MICA and NKG2D illustrates the significance of *MICA* genes and amino acid polymorphisms in biological functions, and this has greatly advanced our understanding of the immune molecule MICA-NKG2D. However, the complex and contradictory nature of tumors suggests that further improvements are needed.

As a stress protein, the main function of MICA on the surfaces of tumor cells is to attract and activate NK cells, enabling them to exert cytotoxic effects. From this perspective, the theory proposed by Luo et al., which states that type I MICA polymorphic molecules have a significantly stronger ability to activate NK cells than type II MICA, can be used to guide the future clinical application of NK cell therapy in patients with various tumors [29]. However, according to Yoshida et al., when NK cells bind to the MICA-129M molecule, although NK cells generate higher levels of IFN-γ, the killing efficiency against K562 cells expressing MICA-129M is somewhat reduced [77]. This may partly explain why some type I MICA molecules can promote tumor progression under certain conditions. Previously, Zou et al. found that MICA*001, MICA*002, MICA*008, and MICA*009 showed significant differences in binding to G1 and G2 antibodies [33]. Luo et al. reported that multiple MICA polymorphic molecules, including MICA*001, MICA*002, MICA*008, and MICA*009, showed significant differences in binding to NKG2D [21]. The α1+α2 domain of the MICA molecule is the binding site for anti-MICA antibodies or NKG2D receptors. The most important polymorphic sites within this region are positions 14, 36, 129, and 173. According to Yang et al., who proposed that amino acid polymorphisms affect the binding energy of MICA-NKG2D [35], it can be inferred that, when the amino acid chain W-C-M-E is present at these four sites, the affinity between MICA and NKG2D is theoretically the strongest. Although this chain pattern does not exist in humans, the W-C-M-K or G-C-M-K amino acid chain of type I MICA proposed by Luo et al. also has high stability and can promote the tight binding of MICA and NKG2D. However, MICA molecules with the Y-V-E amino acid chain at sites 36, 129, and 173 have weaker affinities for NKG2D. From this perspective, the amino acid polymorphisms at specific sites in the α1+α2 domain affect the stability and affinity of MICA-NKG2D, which represents an important molecular basis by which some tumor cells can evade NK cell killing through type II MICA molecules. However, the situation regarding the MICA-NKG2D immune molecule pair is complex. Even type I MICA may not necessarily show strong immune activation. Excessive affinity may lead to the internalization of NKG2D receptors, which is not conducive to the continuous activation of NK cells. However, the molecular mechanism by which excessive affinity between MICA and NKG2D leads to the internalization of NKG2D has not yet been clarified. It may trigger a form of negative feedback within NK cells. See Figure 3 for details.

NKG2D is mainly expressed on the surfaces of NK cells and CD8+ T cells [78]. Moreover, a study has indicated that, when MICA-129M on the cell membrane binds to NKG2D on CD8+ T cells, it can rapidly induce the internalization of NKG2D receptors on the surfaces of CD8+ T cells. However, when MICA-129V binds to CD8+ T cells, it does not significantly reduce the expression density of NKG2D receptors on their surfaces [79]. Unlike NK cells, the function of NKG2D expression on CD8+ T cells is mainly to provide a second signal for CD8+ T cell activation [80]. The activation of CD8+T cells requires two stimulating signals: in the first, the T-cell receptor (TCR) from CD8+T cells binds to HLA-I molecules expressing antigen-presenting cells, while the second signal is provided by co-stimulating molecule pairs. CD28-CD80/86 is the most important second signal for the activation of CD8+T cells, while MICA-NKG2D is another. In the absence of the first or second signal, T cells cannot be activated. When CD8+ T cells lack sufficient CD80 or CD86 stimulation, the ligands on the surfaces of target cells, such as MICA, can replace the CD28-B7 co-stimulatory signals to assist in activating CD8+ T cells. Therefore, the polymorphisms of MICA and other ligands on the surfaces of target cells are essential in enabling CD8+ T cells to exert their killing effects. A study on transplant rejection supports this. The authors found that, in patients undergoing hematopoietic stem cell transplantation, those with the homozygous Met/Met genotype had a significantly higher recurrence rate after transplantation. Further research indicated that, in this group of patients, MICA-129M was expressed on the surfaces of target cells, and it could strongly activate CD8+ T cells in the early stage of transplantation; however, it then caused a decrease in the expression of NKG2D receptors on the surfaces of CD8+ T cells [50]. Without sufficient CD28-B7 co-stimulation, the weakening of the MICA-NKG2D signal can cause a decrease in the activity of CD8+ T cells [81]. The process of high-affinity MICA causing the internalization of NKG2D receptors on the surfaces of CD8+ T cells is shown in Figure 4. NK cells are also affected by the above-described phenomenon. As the most important activating receptor on the surfaces of NK cells, NKG2D shows significant internalization under the strong stimulation of MICA-129M, resulting in the inability of the NKG2D signal to be effectively transmitted to NK cells. This affects the activity of NK cells, weakens the immune surveillance function, and leads to a significant increase in recurrence among patients.

In conclusion, although the M/V polymorphism at the 129th site of the MICA polymorphic molecule significantly affects the intensity of activation of the NKG2D signal, the subsequent negative feedback effect causes NKG2D to be internalized. The body avoids the persistent presence of overly strong NKG2D signals through a negative feedback mechanism. This knowledge supplements the classification of MICA types proposed by Luo et al., confirming that type I MICA can trigger a negative feedback mechanism in the body that is associated with the strength of the MICA-NKG2D signal.

### 4.2. MICA with a C-M-K-G-W-S Amino Acid Pattern Is Responsible for the Internalization of NKG2D, Which Is an Important Factor in the Progression of Cancer

MICA polymorphic molecules with a high affinity for NKG2D can cause the internalization of NKG2D receptors, weakening the killing effects of immune cells such as NK cells and CD8+ T cells. This represents not only a negative feedback mechanism to control the immune response of the body but also an immune escape mechanism for tumor cells, enabling them to evade killing by immune cells such as NK cells. Based on this, it is possible to explain why the prognoses of some tumor patients carrying the *MICA* allele encoding the C-M-K-G-W-S amino acid sequence are worse than those of other patients carrying the MICA allele encoding the Y-V-E-S-R-T amino acid sequence. The report by Ding et al. indicates that the *MICA*012* allele can promote the progression of colon cancer [67]. From the perspective of the NKG2D signal and its negative feedback, MICA*012 has a C-M-K-G-W-S amino acid sequence at the six sites mentioned by Luo et al., representing a MICA molecule that can strongly activate NKG2D and may trigger NKG2D internalization, thereby enabling colon cancer cells to evade killing by immune cells such as NK cells. In addition, Ding et al. note that most colon cancer patients carrying the MICA*012 allele also exhibit *K-ras* gene mutations [67]. This gene is a typical oncogene, and its mutation can promote the proliferation of various tumor cells [82,83,84]. Therefore, under the dual influence of NKG2D internalization and *K-ras* gene mutation, colon cancer cells have the ability to rapidly spread and metastasize. Similarly, Nguyen et al. pointed out that the EB virus copy number in the tumor tissue of nasopharyngeal carcinoma patients carrying the MICA-129M allele was significantly higher than in patients carrying the MICA-129V allele [75]. Although they did not specify the molecular mechanism, EB virus manifests as long-term infection in nasopharyngeal cancer patients, and nasopharyngeal epithelial cells express MICA after stimulation by EB virus. Moreover, patients carrying the MICA-129M allele exhibit the strong activation of NK cells and CD8+ T cells [85,86], but the subsequent negative feedback mechanism leads to the rapid internalization of NKG2D receptors on the surfaces of various immune cells, significantly reducing the intensity of immune surveillance and the killing effect. This may explain the increase in the EB virus copy number, but the specific molecular mechanism still needs further research. According to the report by Antje et al., tumor cells derived from various epithelial cell sources can all produce sMICA [87,88,89], and other soluble NKG2D ligands, such as sMICB [90,91], can even be produced.

Leukemia cells and melanoma cells, which do not originate from epithelial cells, can also produce various soluble NKG2D ligands. Regardless of whether the malignant tumor cells originate from epithelial cells or non-epithelial cells, the mechanism by which soluble NKG2D ligands are generated is similar. The disintegrin-like metalloproteinase (ADAM) family generates soluble products [92,93,94] by cleaving the transmembrane regions of MICA/MICB on the cell membrane. Multiple myeloma cells promote the expression of ADAMs by abnormally activating signaling pathways such as MAPK, thereby further increasing the levels of soluble NKG2D ligands [95]. When these soluble NKG2D ligands bind to the NKG2D receptors on the surfaces of NK cells, the killing signals mediated by pathways such as PI3K/Akt cannot be activated, thereby reducing the cytotoxicity of NK cells [96]. Additionally, the persistent presence of sMICA and other products leads to an “exhaustion” state among NK cells, manifested as a decreased capacity for proliferation and the reduced secretion of cytokines (such as IFN-γ) [97]; this affects the resistance of tumor cells to NK cells and other NKG2D+ immune cells.

Considering that tumor cells expressing MICA-129M produce higher levels of sMICA [68], this can be considered another negative feedback mechanism employed by the body to target the MICA-NKG2D interaction. These soluble ligands significantly affect the immune activity of NK cells [98], CD8+ T cells [99], NKT cells [100], and others.

It remains unclear why tumor cells expressing MICA-129M produce more sMICA than those expressing MICA-129V. According to existing data, various tumor cells (e.g., liver cancer cells) can produce soluble NKG2D ligands such as sMICA, sMICB, sULBP2, etc. The reason for the generation of these soluble ligands is that the MICA and other molecules on the tumor cell surface are hydrolyzed by multiple highly activated metalloproteinases, and the activity of these metalloproteinases is regulated by hypoxia-inducible factor-1α (HIF-1α) [101,102,103,104]. Most solid tumors exist in a dense hypoxic microenvironment, and the high expression of HIF-1α results in the activation of multiple metalloproteinases. The sites where these enzymes hydrolyze MICA or MICB are located in the α3 domain and transmembrane region [105], suggesting that there may be some specific amino acid polymorphism sites in the α3 domain and transmembrane region of MICA/B. These sites may also be relevant to the linkage imbalance proposed by Luo et al. [29]. Moreover, amino acid polymorphisms at these sites cause some MICA molecules to be hydrolyzed by metalloproteinases, generating soluble products. It is possible that, when the amino acid types at these specific sites are biased towards those that are prone to hydrolyzation, the linkage imbalance matches that of C-M-K-G-W-S. Therefore, MICA molecules that can strongly activate NKG2D are prone to hydrolysis by metalloproteinases and generate soluble ligands. However, it should be noted that most MICA molecules with the C-M-K-G-W-S linkage imbalance are beneficial in promoting and mobilizing immune cells in the body to suppress tumor cells. In the IMGT/HLA database, no amino acid polymorphism sites with C-M-K-G-W-S or Y-V-E-S-R-T linkage imbalances in the transmembrane regions of MICA molecules can be found. This is mainly because the tail lengths of MICA molecules are different, making it difficult to identify universal polymorphism sites. Moreover, not all MICA polymorphic molecules with the C-M-K-G-W-S linkage imbalance contain sites that are prone to hydrolysis. Therefore, the analysis of the interaction strength between MICA and NKG2D is challenging. Further evidence is needed to verify the existence of such polymorphism sites to guide future clinical immunotherapy measures.

## 5. The Polymorphism of MICA Molecules Is a Key Factor That Must Be Considered in Future Immunotherapy Based on the MICA-NKG2D Immune Checkpoint

MICA is the most polymorphic non-classical HLA molecule and is closely related to the susceptibility to or occurrence and development of various diseases. The binding of MICA to NKG2D can be regarded as an immune checkpoint. Appropriate activation strength can enable various immune cells, such as NK cells, CD8+ T cells, NKT cells, and γδT cells, to effectively perform immune surveillance and killing. An overly strong MICA-NKG2D signal can lead to the intense and persistent activation of immune cells such as NK cells, causing damage to the body’s tissues. On the other hand, a weak signal can significantly increase the risk of developing tumors or viral infections. Therefore, regulating the signal strength of the MICA-NKG2D immune checkpoint is of great importance for anti-tumor and anti-viral success and in maintaining immune balance. Although NKG2D has a highly conserved gene among the population, MICA has abundant genetic and amino acid polymorphisms, which inevitably affect the signal strength of MICA-NKG2D.

Activating NK cells and CD8+ T cells is crucial in enhancing anti-tumor immunity. Due to the low expression of HLA-I molecules among tumor cells and the fact that most tumor cells do not express HLA-II molecules, the activation of both CD4+ T cells and CD8+ T cells is negatively affected. However, the activation of NK cells does not require the assistance of HLA molecules, and the low expression of HLA-I molecules also helps in the activation of NK cells [106]. Therefore, promoting the killing of tumor cells by NK cells through activating the MICA-NKG2D signaling pathway is an important immunological strategy to improve the prognosis of various cancer populations. At the same time, without sufficient CD28-B7 signal stimulation, CD8+ T cells can be activated and exert cytotoxic effects through the stimulatory signaling of MICA-NKG2D [107], which is also an important pathway in promoting anti-tumor immunity.

Most MICA polymorphic molecules are classified as MICA-129M (with a strong affinity for NKG2D) or MICA-129V (with a low affinity for NKG2D) based on the M/V polymorphism at the 129th amino acid site. For this reason, when NK cell infusion therapy is administered to tumor patients, the MICA genotype is first determined; this has significant immunological significance in predicting the efficacy of NK cell therapy and adjusting the therapeutic regimen. For tumor patients carrying MICA with a strong affinity for NKG2D, a small-dose NK cell infusion can be given multiple times. While the MICA-129M-type polymorphic molecule can strongly activate NK cells, the body will trigger a negative feedback mechanism of NKG2D internalization in response to the high-intensity MICA-NKG2D signal; therefore, infusing low doses of NK cells multiple times can aid in fully activating the NKG2D signal, thereby achieving a strong anti-tumor immune effect. For tumor patients carrying MICA with a low affinity for NKG2D, a larger number of NK cells can be considered for infusion, because the MICA-129V-type polymorphic molecule does not cause NKG2D internalization. Infusing a large number of NK cells can enable continuous anti-tumor effects, but it should also be noted that some MICA-129V-type MICA molecules (e.g., MICA*008) can detach from the cell surface and generate sMICA, which can severely hinder the biological functioning of NK cells. Therefore, in certain cases, the influence of sMICA should be considered, and plasma exchange should be performed in advance to remove it. Since most sMICA is formed by metalloproteinases (including matrix metalloproteinases and integrin-like metalloproteinases), hydrolysis, which involves inhibiting various metalloproteinases in tumor cells, is also an effective means to prevent or reduce the generation of sMICA [108].

Nonetheless, immunotherapies such as NK cell infusion, the inhibition of metalloproteinases, and the removal of sMICA through plasma exchange are not currently being carried out in clinical practice and remain within the theoretical domain. However, NK cells possess significant advantages, as they do not require activation by HLA molecules, and CD8+ T cells have stronger killing abilities after activation than NK cells. Thus, the above measures are promising approaches for future anti-tumor immunotherapy. Sun et al. designed a new type of CAR-T cell by replacing the front single-chain antibody domain with the extracellular domain of NKG2D and then connecting other CAR-T components, including the CD28 domain. They found that this CAR-T cell could effectively kill liver cancer cells. It recognizes a wider range, including MICA, MICB, and ULBP1-6—overcoming the limitations of traditional CAR-T cells, which can only recognize a single target. Thus, it has strong potential for future application [109]. Moreover, this CAR-T cell retains only the extracellular domain of human NKG2D and has a non-NK internal structure. Therefore, it may not be internalized due to the negative feedback mechanism caused by some MICA polymorphic molecules, enabling it to exert a continuous, strong anti-tumor effect.

## 6. Conclusions

As the main member of the body’s immune surveillance system, the activation of NK cells mainly depends on the stimulation of the MICA-NKG2D signal. Moreover, various immune cells, such as CD8+ T cells, are also regulated by the MICA-NKG2D signal. Although the amino acid sequence of NKG2D is highly conserved in the population, MICA has high polymorphism. Studies on the influence of MICA polymorphism on its affinity with NKG2D have been ongoing for many years, and this immunological phenomenon has potential medical significance for cancers and other diseases. There are significant differences in the affinity of different MICA polymorphic molecules with NKG2D, and the activation intensity of the MICA-NKG2D signaling pathway cannot be considered solely based on the polymorphism of MICA. The immune system’s negative feedback in response to overly strong MICA-NKG2D signals also needs to be considered. Multiple immunological mechanisms have caused the aforementioned contradictory yet unified complex phenomenon. The seemingly contradictory phenomenon hides more complex immunological mechanisms that await further exploration.

## Figures and Tables

**Figure 1 biomolecules-16-00047-f001:**
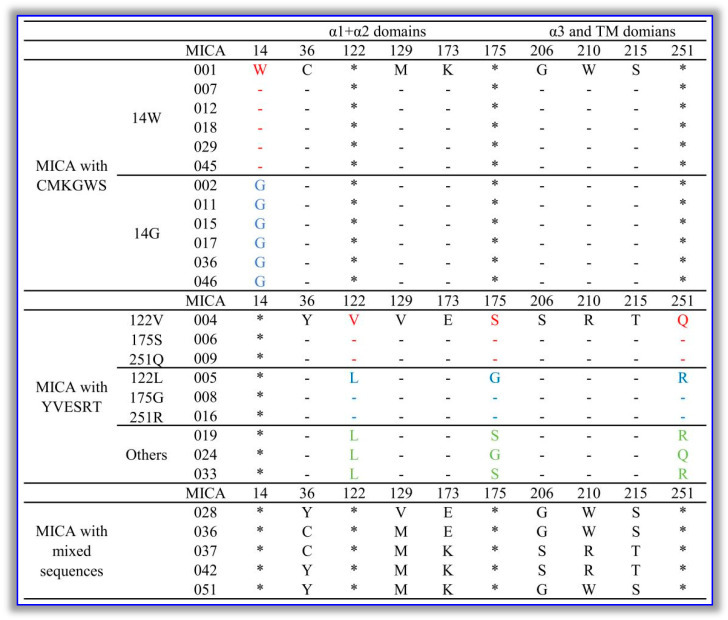
The co-occurrence pattern sites in the amino acid sequences of MICA polymorphic molecules. According to the IMGT/HLA database, all known MICA polymorphic molecules in humans contain amino acid sites with co-occurrence patterns. Based on these amino acid sites, all these molecules can be preliminarily classified into multiple types and subtypes. These sites include positions 14/36/122/129/173/175/206/210/215/251. The symbol “-” indicates consistency with the reference sequence (with MICA*001 and MICA*004 as reference standards, respectively), and the symbol “*” indicates that the polymorphism of this site is not considered.

**Figure 2 biomolecules-16-00047-f002:**
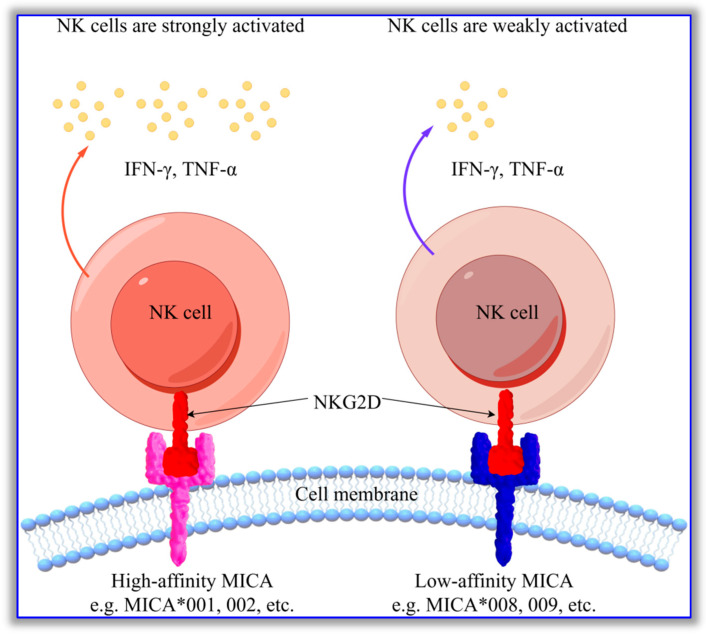
The effect of different affinities of MICA molecules on the activation intensity of NK cells after binding to NKG2D. High-affinity MICA molecules carrying the C-M-K-G-W-S amino acid sequence (including MICA*001, MICA*002, etc.) can strongly activate NK cells and cause them to produce a large amount of cytokines such as IFN-γ and TNF-α after binding to NKG2D, while low-affinity MICA molecules carrying the Y-V-E-S-R-T amino acid sequence (including MICA*008, MICA*009, etc.) have relatively weaker ability to activate NK cells, and the expression of IFN-γ and TNF-α secreted by NK cells is also lower. This diagram was drawn using Figdraw 2.0 software (www.figdraw.com, accessed on 15 December 2025); the ID on the Figdraw website is YIPRYccbef.

**Figure 3 biomolecules-16-00047-f003:**
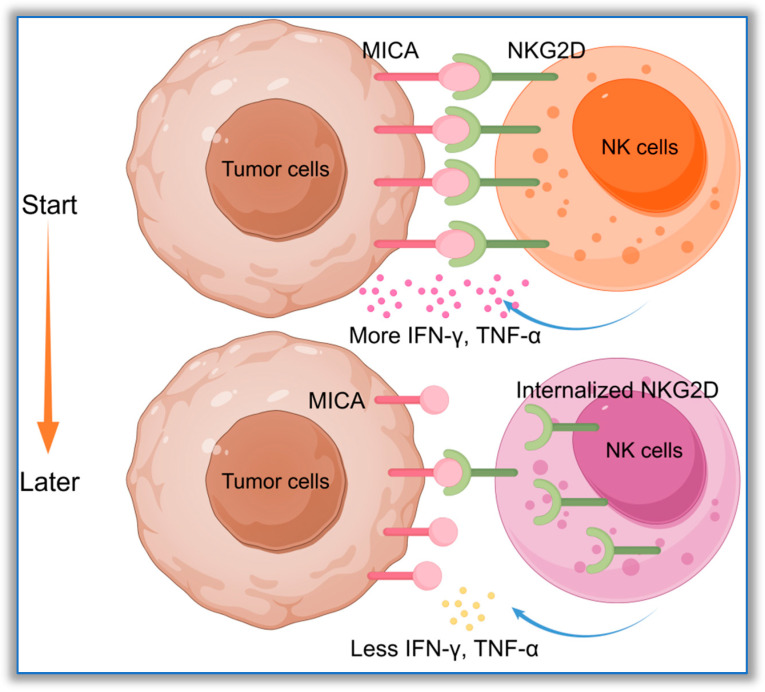
MICA molecules with high affinity for NKG2D mediate the internalization of NKG2D receptors by NK cells. The high-affinity MICA, named Type I MICA, can strongly activate NK cells. The body has developed a negative feedback response to this immune reaction, resulting in the internalization of the NKG2D receptor on the surface of NK cells, weakening the strength of the MICA-NKG2D interaction, and down-regulating the expression of cytokines. This diagram was drawn using Figdraw 2.0 software (www.figdraw.com, accessed on 10 October 2025); the ID on the Figdraw website is RRSWS4d3d5.

**Figure 4 biomolecules-16-00047-f004:**
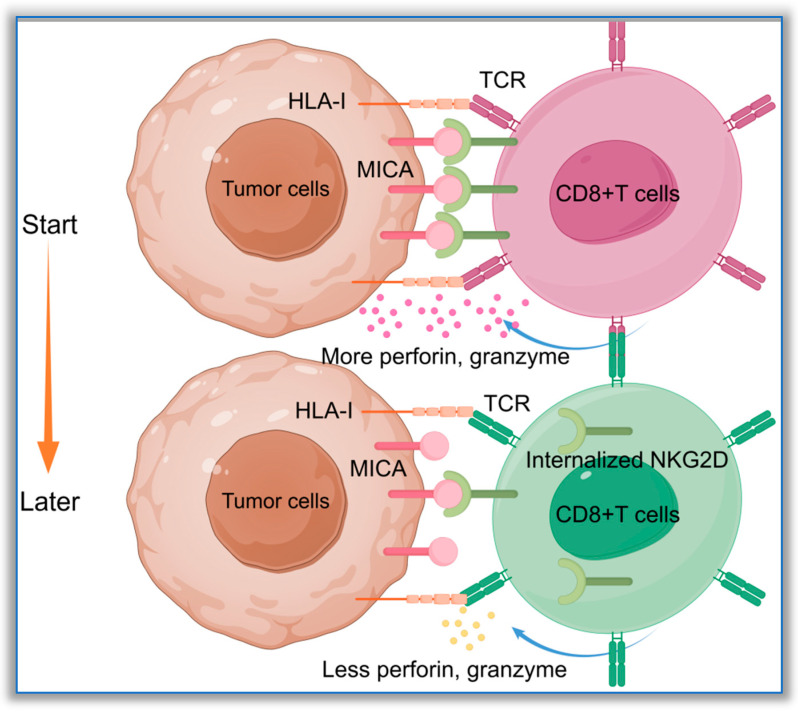
MICA molecules with high affinity for NKG2D mediate the internalization of NKG2D receptors by CD8+ T cells. The MICA expressed by tumor cells can bind to the NKG2D receptor on the surface of CD8+ T cells, providing a second activation signal for CD8+ T cells. After the high-affinity MICA binds to NKG2D, it can strongly activate CD8+ T cells and trigger a negative feedback effect, causing the NKG2D receptors on the surface of CD8+ T cells to be internalized. This diagram was drawn using Figdraw 2.0 software (www.figdraw.com, accessed on 10 October 2025); the ID on the Figdraw website is URRPY87482.

**Table 1 biomolecules-16-00047-t001:** Common cancers expressing NKG2D ligands.

Cancer	Type of NKG2D Ligands	References
Lung Cancer	MICA, MICB, ULBPs	Zhi [33], Kim [34]
Liver Cancer	MICA, MICB, ULBP2	Chen [35], Weng [36]
Gastric Cancer	MICA, MICB, ULBP2	Ribeiro [37], Zhang [38]
Colorectal Cancer	MICA, MICB, ULBPs	Wang [39], McGilvray [40]
Nasopharyngeal Carcinoma	MICA	Ben Chaaben [41]
Cervical Cancer	MICA, MICB, ULBP1	Cho [42]
Breast Cancer	MICA, ULBP2	Wang [43], Kaidun [44]
Cholangiocarcinoma	MICA, MICB, ULBPs	Tsukagoshi [45]
Pancreatic Cancer	MICA, MICB	Duan [46]
Acute Myeloid Leukemia	MICA, MICB, ULBPs	Alves da Silva [47], Story [48]
Chronic Myeloid Leukemia	MICA, ULBP2	Closa [49], Cheon [50]
Acute Lymphocytic Leukemia	MICA, MICB	Li [51]
Chronic Lymphocytic Leukemia	MICA, ULBP3	Zhang [52], Poggi [53]
Melanoma	MICA, MICB	Di Donato [54], Bilotta [55]

Note: The ULBP family consists of 6 members, namely ULBP1–6. ULBPs in table represent various ULBP molecules.

## Data Availability

No new data were created or analyzed in this study.

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
