# Peer review of "Biomolecules2026, 16(1), 47;https://doi.org/10.3390/biom16010047"

_biomolecules, 2025, doi:10.3390/biom16010047_

Round 1

Reviewer 1 Report

Comments and Suggestions for Authors

To improve our understanding of immune surveillance and tumor immunity, it is crucial to better understand how MICA amino acid polymorphisms affect the MICA-NKG2D interaction. The submitted review by Chuyu Xiao et al. attempts to address this very issue.

While the submitted review covers important aspects of this subject, the manuscript it is not yet ready for publication. Overall, it requires substantial editing and improvements to its content and structure.

The English usage is often not up to publication quality, with awkward sentences scattered throughout the manuscript. Representative paragraphs can be found at lines 418, 452, 492, and 521.

This review primarily focuses on MICA, though the authors note that other antigens, including MICB and ULBP family members, also bind to NKG2D. However, details on MICB are limited in the early sections and are explained more clearly later for both MICB and ULBP. To improve clarity, the authors should remove paragraphs 5.1 and 5.2 and incorporate their key points into the introduction, making it clear that MICA is the central focus of the review.

The authors correctly state that MICA is primarily expressed in tumor cells of epithelial origin. However, it is important to mention that some non-epithelial tumors, such as melanomas and certain types of leukemia, also express MICA. The authors could include a table summarizing this information.

Regarding leukemia (paragraph 3.1, line 322), the authors should specify the types of leukemia that express MICA. They should also dedicate more time to describing how leukemia cells protect themselves from NK cell-mediated cytotoxicity by shedding MICA and MICB.

The authors should provide a more in-depth analysis of the functional differences between type I and type II MICA variants and their roles in immune evasion. It would also be valuable to clarify how the distinct NKG2D binding affinities and signaling strengths of these MICA types influence the activation or exhaustion of NK and CD8+ T cells.

Comments on the Quality of English Language

The English usage is often not up to publication quality, with awkward sentences scattered throughout the manuscript (see comment to authors for more details).

Author Response

Comment 1: The English usage is often not up to publication quality, with awkward sentences scattered throughout the manuscript. Representative paragraphs can be found at lines 418, 452, 492, and 521.

Response 1: Thank you very much for your suggestion. We have sent this manuscript to the professional staff of Biomolecules Journal for English revision, and have made comprehensive corrections to the words and grammar in our resubmitted manuscript.

Comment 2: This review primarily focuses on MICA, though the authors note that other antigens, including MICB and ULBP family members, also bind to NKG2D. However, details on MICB are limited in the early sections and are explained more clearly later for both MICB and ULBP. To improve clarity, the authors should remove paragraphs 5.1 and 5.2 and incorporate their key points into the introduction, making it clear that MICA is the central focus of the review.

Response 2: Thank you very much for your suggestion. We have removed the contents of Sections 5.1 and 5.2 from the original manuscript and simplified them, incorporating them into the preface. Please refer to lines 63-77 for details.

Comment 3: The authors correctly state that MICA is primarily expressed in tumor cells of epithelial origin. However, it is important to mention that some non-epithelial tumors, such as melanomas and certain types of leukemia, also express MICA. The authors could include a table summarizing this information.

Response 3: Thank you very much for your correct suggestion. We have made a list of tumors expressing MICA or other NKG2D ligands for easier reading by readers. Please refer to Table 1 (lines 110-111) for details.

Comment 4: Regarding leukemia (paragraph 3.1, line 322), the authors should specify the types of leukemia that express MICA. They should also dedicate more time to describing how leukemia cells protect themselves from NK cell-mediated cytotoxicity by shedding MICA and MICB.

Response 4: Thank you very much for your valuable suggestions. We have described the types of leukemia cited in the references in lines 322-325, and have added relevant contents in lines 519-533 regarding the generation of soluble MICA/B by leukemia or other tumors.

Comment 5: The authors should provide a more in-depth analysis of the functional differences between type I and type II MICA variants and their roles in immune evasion. It would also be valuable to clarify how the distinct NKG2D binding affinities and signaling strengths of these MICA types influence the activation or exhaustion of NK and CD8+ T cells.

Response 5: Thank you for your valuable suggestions. We have expanded this part of the content. Relevant additions have been made in lines 269-284, 377-386, and 404-428 of the manuscript.

Reviewer 2 Report

Comments and Suggestions for Authors

The authors address an important and timely topic concerning MICA polymorphisms, their impact on the MICA-NKG2D immunological axis, and potential implications for cancer immunotherapy. The manuscript provides an extensive literature overview and discusses complex immunobiological mechanisms. However, the current version requires substantial revision, as it contains numerous structural and interpretational issues that limit its clarity and overall scientific quality.

Major comments:

Introduction

  1. The Introduction is scientifically rich but would benefit from a clearer and more concise structure. It should primarily establish the rationale, define the knowledge gap, and state the objectives of the review. I recommend reducing repetition and sharpening the focus on what the manuscript aims to address.
  2. Several sentences contain grammatical errors, awkward phrasing, or typographical issues. For example, Line 83 includes the incorrect construction: “has been has not been fully clarified”. A thorough language revision is necessary.
  3. The statement in Line 54 incorrectly categorises MICA/B as HLA class III genes. MICA and MICB are non-classical class I molecules and should not be grouped within HLA class III. Please revise to reflect the current immunogenetic classification.

Section 2.

  1. This section appears to largely reproduce data from the IMGT/HLA database without sufficient selection of information relevant to the central hypotheses. The reader may lose track of the main biological question and the functional significance of the described polymorphisms. Please focus on variants with demonstrated functional effects rather than listing all known substitutions.
  2. The repeated use of long amino acid strings (e.g., C-M-K-G-W-S) significantly hinders readability. Presenting these data in a table, figure, or structural context would make the information more accessible.
  3. Although the topic is valuable, the section is overly detailed and difficult to follow. Studies by Steinle, Zou, Luo, and others are summarised without sufficient discussion of methodological limitations or differences. A clearer synthesis of their contributions is needed to justify the scientific relevance of the described polymorphisms.
  4. The term „linkage disequilibrium” is used imprecisely. The described observations represent co-occurrence patterns in protein sequences rather than statistically demonstrated LD in human populations. Without population-level data (D’, r2), please replace this terminology (e.g., co-occurrence patterns, haplotype clusters) or add proper LD analysis.

Section 3.

  1. Results from different cancer types and study designs are presented together, but the section lacks a critical synthesis explaining why the effects of specific MICA variants may differ across clinical contexts.
  2. The manuscript does not clearly distinguish between membrane-bound MICA and soluble MICA (sMICA), which have opposing immunological consequences. These heterogeneous findings (in vitro, clinical, viral contexts) should be organised into a coherent model explaining how the same allele can exert both protective and detrimental effects depending on its form and regulation.

Section 4.

  1. The manuscript does not sufficiently distinguish the different biological roles of NKG2D on NK cells vs CD8⁺ T cells, and some conclusions are inappropriately extrapolated between these cell types. A clearer separation of mechanisms is needed.
  2. There is substantial redundancy between the text and the descriptions of Figures 3 and 4. I recommend reducing duplicated content and focusing on interpretation rather than reiterating the details of figures.
  3. The section blends established mechanisms (e.g., HIF-1α regulation of metalloproteinases, shedding of MICA/MICB) with isolated study findings and author interpretations without distinguishing between confirmed data and proposed models.

Section 5.

  1. This Section introduces an important topic by addressing other NKG2D ligands; however, the current discussion is very brief and lacks sufficient depth to support meaningful conclusions. I recommend expanding (or alternatively removing) this section to provide a more comprehensive and structured overview of MICB and ULBPs, including their known expression patterns and disease associations.

Section 6.

  1. This part proposes immunotherapeutic strategies based on MICA polymorphisms. However, several recommendations are presented too definitively given the current level of evidence (e.g., Lines 579–598). The suggested clinical approaches, such as tailoring NK cell infusion regimens to MICA-129 genotype, plasma exchange to remove sMICA, or metalloproteinase inhibition, remain hypothetical and should be more clearly framed as future directions requiring validation. A clearer distinction between established knowledge and speculative therapeutic implications would strengthen the credibility of this section.

Conclusions

  1. The Conclusions are very general and do not clearly summarise the key insights.
  2. The contradictions and complexity highlighted in Sections 3 and 4 (e.g., dual roles of high-affinity variants, influence of shedding and immune regulation) should be acknowledged in the Conclusions, as they represent an important message of the review.

Author Response

Comment 1: Introduction

The Introduction is scientifically rich but would benefit from a clearer and more concise structure. It should primarily establish the rationale, define the knowledge gap, and state the objectives of the review. I recommend reducing repetition and sharpening the focus on what the manuscript aims to address.

Response 1: Thank you very much for your suggestion. We have revised the relevant content. However, since the original 5.1 and 5.2 content has been condensed and placed in the preface, and a table showing the expression of MICA and other ligands by tumor cells has been added, the length of the document may still be quite long. Nevertheless, we have indeed refined the text content.

Comment 2: Several sentences contain grammatical errors, awkward phrasing, or typographical issues. For example, Line 83 includes the incorrect construction: “has been has not been fully clarified”. A thorough language revision is necessary.

Response 2: Thank you very much for your suggestion. We have sent this manuscript to the professional staff of Biomolecules Journal for English revision, and have made comprehensive corrections to the words and grammar in our resubmitted manuscript.

Comment 3: The statement in Line 54 incorrectly categorises MICA/B as HLA class III genes. MICA and MICB are non-classical class I molecules and should not be grouped within HLA class III. Please revise to reflect the current immunogenetic classification.

Response 3: Thank you for your correct guidance. We have already removed this incorrect statement.

Comment 4: Section 2

This section appears to largely reproduce data from the IMGT/HLA database without sufficient selection of information relevant to the central hypotheses. The reader may lose track of the main biological question and the functional significance of the described polymorphisms. Please focus on variants with demonstrated functional effects rather than listing all known substitutions.

Response 4: Thank you very much for your correct guidance. We have revised this part and focused on describing the amino acid polymorphism sites that are relevant to the subsequent content of this article. We have removed the simple listing of all the variant sites.

Comment 5: The repeated use of long amino acid strings (e.g., C-M-K-G-W-S) significantly hinders readability. Presenting these data in a table, figure, or structural context would make the information more accessible.

Response 5: Thank you very much for your suggestion. As the MICA amino acid polymorphism sites mentioned in our manuscript are closely related to the affinity between MICA and NKG2D, as well as the activation degree of immune cells such as NK cells, it is inevitable to list the relevant polymorphism sites. However, we have removed the frequently occurring long strings in some parts. Please understand and forgive us. Thank you again.

Comment 6: Although the topic is valuable, the section is overly detailed and difficult to follow. Studies by Steinle, Zou, Luo, and others are summarised without sufficient discussion of methodological limitations or differences. A clearer synthesis of their contributions is needed to justify the scientific relevance of the described polymorphisms.

Response 6: Thank you very much for your suggestion. We did overlook this important content. We have added relevant content in lines 269-284, and comprehensively analyzed the differences and limitations of their research.

Comment 7: The term linkage disequilibrium” is used imprecisely. The described observations represent co-occurrence patterns in protein sequences rather than statistically demonstrated LD in human populations. Without population-level data (D’, r2), please replace this terminology (e.g., co-occurrence patterns, haplotype clusters) or add proper LD analysis.

Response 7: Thank you very much for your valuable guidance. The term "linkage disequilibrium" is indeed not accurate in our manuscript. We have replaced "linkage disequilibrium" with "co-expression pattern".

Comment 8: Section 3

Results from different cancer types and study designs are presented together, but the section lacks a critical synthesis explaining why the effects of specific MICA variants may differ across clinical contexts. The manuscript does not clearly distinguish between membrane-bound MICA and soluble MICA (sMICA), which have opposing immunological consequences. These heterogeneous findings (in vitro, clinical, viral contexts) should be organised into a coherent model explaining how the same allele can exert both protective and detrimental effects depending on its form and regulation.

Response 8: Thank you very much for your valuable guidance. We have made comprehensive revisions to this part. Firstly, we integrated the content describing various solid tumors in 3.1, eliminating repetitive content and conducting a comprehensive analysis. Secondly, in the last paragraph of 3.2, we described the soluble NKG2D ligand, clearly indicating that this form of MICA has the biological effect of inducing tumor immune escape. Please refer to lines 269-284 and 377-386 for details.

Comment 9: Section 4

The manuscript does not sufficiently distinguish the different biological roles of NKG2D on NK cells vs CD8⁺ T cells, and some conclusions are inappropriately extrapolated between these cell types. A clearer separation of mechanisms is needed.

Response 9: Thank you very much for your valuable suggestions. We added the functional differences of NKG2D in NK cells and CD8+ T cells in lines 404-428 and 444-449. We focused on describing that the NKG2D receptor on the surface of NK cells is an important activating receptor, while the main function of the NKG2D receptor on the surface of CD8+ T cells is to provide a second stimulatory signal for the activation of T cells.

Comment 10: There is substantial redundancy between the text and the descriptions of Figures 3 and 4. I recommend reducing duplicated content and focusing on interpretation rather than reiterating the details of figures. The section blends established mechanisms (e.g., HIF-1α regulation of metalloproteinases, shedding of MICA/MICB) with isolated study findings and author interpretations without distinguishing between confirmed data and proposed models.

Response 10: Thank you very much for your suggestion, we have already reduced the repetitive content.

Comment 11: Section 5

This Section introduces an important topic by addressing other NKG2D ligands; however, the current discussion is very brief and lacks sufficient depth to support meaningful conclusions. I recommend expanding (or alternatively removing) this section to provide a more comprehensive and structured overview of MICB and ULBPs, including their known expression patterns and disease associations.

Response 11: Thank you very much for your suggestion. Other reviewers also brought this up, and we hope to condense this section. Therefore, we have revised the relevant content.

Comment 12: Section 6.

This part proposes immunotherapeutic strategies based on MICA polymorphisms. However, several recommendations are presented too definitively given the current level of evidence (e.g., Lines 579–598). The suggested clinical approaches, such as tailoring NK cell infusion regimens to MICA-129 genotype, plasma exchange to remove sMICA, or metalloproteinase inhibition, remain hypothetical and should be more clearly framed as future directions requiring validation. A clearer distinction between established knowledge and speculative therapeutic implications would strengthen the credibility of this section.

Response 12: Thank you very much for your correct guidance. We have added new relevant contents in lines 616-622, indicating that these ways such as NK cell infusion, continuous metalloproteinase, and removal of sMICA from plasma are still within the realm of theoretical speculation. We also pointed out that these methods have potential value in future clinical applications.

Comment 13: Conclusions

The Conclusions are very general and do not clearly summarise the key insights.

The contradictions and complexity highlighted in Sections 3 and 4 (e.g., dual roles of high-affinity variants, influence of shedding and immune regulation) should be acknowledged in the Conclusions, as they represent an important message of the review.ge and speculative therapeutic implications would strengthen the credibility of this section.

Response 13: Thank you very much for your guidance. We have rewritten the contents of the conclusion section, emphasizing the seemingly contradictory phenomenon between the affinity of MICA and NKG2D and the activation degree of immune cells such as NK cells. We also propose that behind this seemingly contradictory phenomenon lie more complex immunological mechanisms, which await further exploration.

Reviewer 3 Report

Comments and Suggestions for Authors

I would like to thank you for providing a comprehensive and well-organized summary of MICA protein amino acid polymorphisms and the clinical implications of major findings. The paper has a clear structure and is understandable even to a specialist who does not work narrowly in this field. Below, I offer some suggestions that may help to improve your manuscript and make it more prominent.

  1. I recommend reviewing the text for grammatical issues, inconsistencies, or missing words. For example, lines 389-390, 492, 599-601, 614). I also suggest avoiding excessively long sentences, which are difficult to understand or analyze (e.g. lines 201-205). 
  2. I suggest to transfer the list of cells expressing NKG2D receptors (now line 89) to the point when this receptor is first characterized (lines 78-79). 
  3. In the text, you state that there are three types of MICA proteins, based on their linkage disequilibrium states, and ask to refer to Fig. 1 (lines 155-156). However, Fig. 1 shows only two major groups. You also describe a number of intriguing MICA proteins of the third group that share polymorphisms from the others (lines 245-248). My suggestion is to add these MICA protein (MICA*028, MICA*037 and MICA*042) sequences to Fig.1, as the mentioned third group. I believe that this will make the comparison of their sequences easier and the reference (lines 155-156) accurate. From the case of MICA*042, do you think that the amino acid at the 36th position is crucial for binding strength? 
  4. Lines 199-209: according to the text, both MICA*001 and MICA*002 have M in the 129th position and bind to G2 antibody, and both MICA*008 and MICA*009 - have V and bind to G1 antibody. You note that there is an ‘opposition phenomenon’’, but I (as a reader, I do not clearly see the inconsistency you refer to. 
  5. There are several iterations of the same information in the text. Though in some cases it helps to refresh the main theses of the paper, in others it creates an impression of reading the same. For example, in section 3.1 the 3 first paragraphs illustrate the same idea: Type-I MICA are more potent activators of NK and CD8+ T cells, thus patients which obtain this phenotype are more defended against tumor progression, - and vice versa. In my opinion, you could reorganize the position of main thoughts, not to repeat them multiple times.
  6. Why in some cases strong binding leads to the internalization of NKG2D receptors, and in others - don’t? In the article you differentiate a subcategory for such type MICA molecules with high affinity, but do not give any suggestions on the mechanisms of why this happens. I think that it would be beneficial to add some suggestions, if there are any in the literature.
  7. Line 416: it may help to briefly explain what is meant by the “second activation signal” (and what constitutes the first). Lines 434-436: I suggest moving the reference to Fig. 4 up to where CD8+ T cells are described (lines before 429).

Author Response

Comment 1: I recommend reviewing the text for grammatical issues, inconsistencies, or missing words. For example, lines 389-390, 492, 599-601, 614). I also suggest avoiding excessively long sentences, which are difficult to understand or analyze (e.g. lines 201-205).

Response 1: Thank you very much for your suggestion. We have sent this manuscript to the professional staff of Biomolecules Journal for English revision, and have made comprehensive corrections to the words and grammar in our resubmitted manuscript.

Comment 2: I suggest to transfer the list of cells expressing NKG2D receptors (now line 89) to the point when this receptor is first characterized (lines 78-79).

Response 2: Thank you for your meticulous observation. We have moved this paragraph to lines 80-81 of the current manuscript.

Comment 3: In the text, you state that there are three types of MICA proteins, based on their linkage disequilibrium states, and ask to refer to Fig. 1 (lines 155-156). However, Fig. 1 shows only two major groups. You also describe a number of intriguing MICA proteins of the third group that share polymorphisms from the others (lines 245-248). My suggestion is to add these MICA protein (MICA*028, MICA*037 and MICA*042) sequences to Fig.1, as the mentioned third group. I believe that this will make the comparison of their sequences easier and the reference (lines 155-156) accurate.

Response 3: Thank you for your meticulous observation. We have revised Figure 1 according to your suggestions and added relevant contents.

Comment 4: From the case of MICA*042, do you think that the amino acid at the 36th position is crucial for binding strength?

Response 4: Thank you very much for your valuable guidance. For MICA*042, there are no relevant references indicating that the polymorphism of the 36th amino acid has an impact on its affinity with NKG2D now. In our manuscript, we can only provide a brief description of the phenomenon. Due to the lack of more proofs, we were unable to discuss the specific biological function of this position.

Comment 5: Lines 199-209: according to the text, both MICA*001 and MICA*002 have M in the 129th position and bind to G2 antibody, and both MICA*008 and MICA*009 - have V and bind to G1 antibody. You note that there is an ‘opposition phenomenon’’, but I (as a reader, I do not clearly see the inconsistency you refer to.

Response 5: Thank you very much for your suggestion. We have described this opposing phenomenon in lines 203-205 of our new manuscript. The opposing phenomenon we mentioned is that the binding strength of MICA*001/002 or MICA*008/009 to G1 or G2 antibodies shows opposite characteristics. For example, the mean fluorescence intensity (MFI) of MICA*002 with G1 antibody is >5000, while with G2 antibody it is <1000.

Comment 6: There are several iterations of the same information in the text. Though in some cases it helps to refresh the main theses of the paper, in others it creates an impression of reading the same. For example, in section 3.1 the 3 first paragraphs illustrate the same idea: Type-I MICA are more potent activators of NK and CD8+ T cells, thus patients which obtain this phenotype are more defended against tumor progression, - and vice versa. In my opinion, you could reorganize the position of main thoughts, not to repeat them multiple times.

Response 6: Thank you very much for your suggestion. We have rewritten this part and removed the repetitive content.

Comment 7: Why in some cases strong binding leads to the internalization of NKG2D receptors, and in others - don’t? In the article you differentiate a subcategory for such type MICA molecules with high affinity, but do not give any suggestions on the mechanisms of why this happens. I think that it would be beneficial to add some suggestions, if there are any in the literature.

Response 7: Thank you very much for your valuable suggestions. As there are currently no relevant references explaining the specific mechanism between the affinity of MICA-NKG2D and the internalization of NKG2D receptors, we can only temporarily attribute this phenomenon to a negative feedback regulation of the immune system at present.

Comment 8: Line 416: it may help to briefly explain what is meant by the “second activation signal” (and what constitutes the first).

Response 8: Thank you very much for your guidance. We have added the contents about the first and second signals of T cell activation in lines 445-450 of the new manuscript.

Comment 9: Lines 434-436: I suggest moving the reference to Fig. 4 up to where CD8+ T cells are described (lines before 429).

Response 9: Thank you very much for your guidance. We have moved this content to the position you suggested (See Line 463-464).

Round 2

Reviewer 2 Report

Comments and Suggestions for Authors

The manuscript has been substantially improved compared with the previous version. The remaining issues are minor and mainly relate to terminology consistency.

  1. The manuscript uses the non-standard term “W/G di-stability” (e.g., lines 343 and 348) when referring to residue 14 and its potential impact on MICA-NKG2D binding. Please replace this with standard terminology, such as “W/G amino-acid polymorphism (dimorphism) at position 14.” If the authors intend to discuss functional/energetic implications, please phrase this explicitly as “variant-dependent differences in binding affinity/complex stability” (rather than “di-stability”).
  2. Lines 327-328, 342 and 351: “co-expression pattern” is used to describe residue combinations co-occurring across sequence positions (36/129/173/206/210/215) within MICA alleles. As “co-expression” usually refers to gene/protein expression, authors should replace it with “co-occurrence pattern” or “sequence/residue pattern” for clarity.

Author Response

Comment 1: The manuscript uses the non-standard term “W/G di-stability” (e.g., lines 343 and 348) when referring to residue 14 and its potential impact on MICA-NKG2D binding. Please replace this with standard terminology, such as “W/G amino-acid polymorphism (dimorphism) at position 14.” If the authors intend to discuss functional/energetic implications, please phrase this explicitly as “variant-dependent differences in binding affinity/complex stability” (rather than “di-stability”).

Response 1: Thank you very much for your suggestion. We have replaced the non-standard term "W/G di-stability" with "W/G dimorphism". See the lines 343 and 348.

Comment 2: Lines 327-328, 342 and 351: “co-expression pattern” is used to describe residue combinations co-occurring across sequence positions (36/129/173/206/210/215) within MICA alleles. As “co-expression” usually refers to gene/protein expression, authors should replace it with “co-occurrence pattern” or “sequence/residue pattern” for clarity.

Response 2: Thank you very much for your suggestion. We have replaced all occurrences of "co-expression" in the manuscript with "co-occurrence". See the lines 125,131,133,134,135,136,145,150,154,156,161,164,170,173,243,245,271,294,327,342 and 351.

Reviewer 3 Report

Comments and Suggestions for Authors

Dear authors,

thank you very much for the positive attitude to the comments and suggestions. I am especially thankful for bringing changes to Figure 1.

The manuscript represents high quality and is more clear for analysis.

Author Response

Comment 1: thank you very much for the positive attitude to the comments and suggestions. I am especially thankful for bringing changes to Figure 1. The manuscript represents high quality and is more clear for analysis.

Response 1: Thank you very much for your recognition. We have also learned a lot from your meticulous guidance. Once again, we would like to express our gratitude for your hard work and professional guidance.